# Towards an AI-Driven Data Reduction Framework for Smart City Applications

**DOI:** 10.3390/s24020358

**Published:** 2024-01-07

**Authors:** Laercio Pioli, Douglas D. J. de Macedo, Daniel G. Costa, Mario A. R. Dantas

**Affiliations:** 1INE, Computer Science Department, Federal University of Santa Catarina, Florianopolis 88040-370, Brazil; laercio.pioli@posgrad.ufsc.br (L.P.); douglas.macedo@ufsc.br (D.D.J.d.M.); 2Department of Information Science, Federal University of Santa Catarina, Florianopolis 88040-370, Brazil; 3INEGI, Faculty of Engineering, University of Porto, 4169-007 Porto, Portugal; 4ICE, Computer Science Department, Federal University of Juiz de Fora, Juiz de Fora 36036-900, Brazil; mario.dantas@ice.ufjf.br

**Keywords:** Internet of Things, artificial intelligence, edge intelligence, machine learning, urban sensing

## Abstract

The accelerated development of technologies within the Internet of Things landscape has led to an exponential boost in the volume of heterogeneous data generated by interconnected sensors, particularly in scenarios with multiple data sources as in smart cities. Transferring, processing, and storing a vast amount of sensed data poses significant challenges for Internet of Things systems. In this sense, data reduction techniques based on artificial intelligence have emerged as promising solutions to address these challenges, alleviating the burden on the required storage, bandwidth, and computational resources. This article proposes a framework that exploits the concept of data reduction to decrease the amount of heterogeneous data in certain applications. A machine learning model that predicts a distortion rate and its corresponding reduction rate of the imputed data is also proposed, which uses the predicted values to select, among many reduction techniques, the most suitable approach. To support such a decision, the model also considers the context of the data producer that dictates the class of reduction algorithm that is allowed to be applied to the input stream. The achieved results indicate that the Huffman algorithm performed better considering the reduction of time-series data, with significant potential applications for smart city scenarios.

## 1. Introduction

The exponential growth in data production is placing a significant strain on computing elements, encompassing storage, network infrastructure, processing power, and security [1,2,3]. This is primarily due to massive heterogeneous data from multiple sources in the age of the Internet of Things (IoT), which is particularly significant for many smart city applications [4,5]. As they are mostly expected to be comprised a myriad of smart urban services [6], with data being generated by sensors, wearable devices, connected vehicles, social media, and external systems, the resulting scenario will be defined by the generation of huge amounts of data, on a daily basis [7,8]. In order to address the potential systems degradation resulting when handling the processing, storage, and distribution of such data, efficient mechanisms to assure data reduction (DR) are highly necessary, fostering investigation efforts in this area.

Preprocessing the produced data prior to its transmission to servers emerges as a promising solution to tackle data production issues. This preliminary processing step plays a crucial role in the context of IoT due to the heterogeneous and often large-scale data production. This processing handles unstructured, semi-structured, and structured data, making them available to be stored and used by applications. Hence, ensuring data quality in sensing environments that can be reached by introducing data preprocessing techniques becomes indispensable [9,10].

Efficient data management in sensor-based applications is a complex and challenging task, especially when they are required in real-time applications [11]. Unpredictable parameters associated with sensor data make this process even more difficult. Actually, such required efficiency is relevant for the network’s architectural configuration, which is particularly critical in wireless sensor networks, an important component of many IoT-based applications. Therefore, the limited availability of processing, memory, and energy resources in such networks has demanded intelligent data management to increase their usability in a lot of emerging monitoring scenarios.

In IoT systems, DR techniques such as data compression (DC), data filtering, and data aggregation are commonly utilized when trying to ensure that applications handle only relevant data. However, the adoption of such DR techniques may significantly impact system performance due to their associated computational costs, with important differences in introducing DR techniques according to architectural IoT layers. Introducing it closer to their data production, for example, into sensors, or even edge devices, might bring advantages compared to approaches that implement it far from its production. In the first place, network performance might be increased, usually reducing network bandwidth latency and avoiding congestion at the gateway. On the other hand, introducing it as far as possible from its origin requires transmitting raw data to the cloud, leading to network congestion due to large-scale data transmission. Therefore, to maintain system performance and prevent delays caused by transmitting large data loads to cloud servers, it is essential to implement DR techniques closer to the data production sources [12].

Many computational advantages can be reached by introducing DR techniques to handle raw data at the edge of the network; network latency reductions, computational storage savings, energy cost reductions, and device lifetime increases are just some of them. Bandwidth network latency in the gateway also can be reduced, thereby eliminating I/O bottlenecks in network connections [1]. It usually happens due to the fact that these components are overused with raw data injection. In this context, many works have proposed DR solutions operating at the edge of IoT systems with the aim of reducing the total amount of transmitted data, with varying performance levels [13,14,15,16,17,18,19,20]. With increasing complexities in this area, however, new promising solutions are highly welcome.

In a different perspective, DR could be achieved by leveraging Artificial Intelligence (AI) mechanisms. When doing so, there would be many ways to achieve DR, which could be (i) directly applied closer to the data source, (ii) introduced on the fog layer that has more computing power than edge devices, or (iii) in the cloud, where computational processing power is usually superior to the edge. All of these configurations might bring advantages and drawbacks to the applications as a whole, particularly for smart city scenarios, although moving processing closer to the edge may not be straightforward.

This article evaluates relevant parameters and proposes a new method that focuses on data volume reduction while preserving data quality, potentially benefiting IoT-based smart city applications in multiple scenarios. A multi-tier framework that can handle (huge amounts of) heterogeneous data without compromising data integrity or meaning is proposed, exploiting both data clustering and Machine Learning (ML) inferences. Such a combined solution for the defined scenario has not been proposed before, to the best of our knowledge.

The contributions of this article are threefold and defined as follows:A proposal to perform optimized DR for heterogeneous IoT sources in real-time, acting closer to the physical world to decreasing bandwidth demands and overall energy consumption;Definition of an innovative multi-tier framework that exploits different AI algorithms to fine-tune DR algorithms, achieving higher efficiency;Support the development of IoT-based applications by the adoption of a reference sensor-edge-fog framework, particularly targeted at DR for smart city scenarios;

This article is organized as follows. Section 2 discusses related works, whereas Section 3 presents the proposed framework, including its definition for each architectural layer. The experimental configurations, encompassing contexts, scenarios, environment, algorithms, and datasets, are outlined in Section 4. Section 5 presents the achieved evaluation results and envisioned applications in this area. Finally, Section 6 concludes this article and outlines some future steps.

## 2. Related Works

This section introduces the current state-of-the-art and outlines the key research points that have influenced our work in multiple ways. Numerous authors have investigated DR frameworks that operate at the edge of IoT systems intending to decrease the total transmitted data volume. The discussed papers are then some important contributions in this broad research area.

In [14], the authors introduced a data handler framework with a primary objective of data volume reduction. Their approach involved a thorough analysis of established DR methods, including techniques like data sampling, piecewise approximation, selective forwarding, Perceptually Important Points (PIP), and change detection. Additionally, the authors proposed the creation of three algorithms specifically designed to facilitate the real-time reduction of time series input elements, all of which are founded on the concept of PIP. Overall, this concept proved to be a key factor in achieving effective data volume reduction at the edge.

Incorporating AI at the edge of IoT systems to enhance DR efforts has also been observed. In [21], authors proposed a method based on Actor-Critic with Temporal Difference Off-Policy (ATOC) reinforcement learning. This method was designed to make decisions regarding the forwarding or dropping of sensor data to reduce communication overhead on the edge. The solution involves a controller, specifically implemented as a neural network, that governs data flow based on sensor information. Initially, sensing data are sent to an edge module, which evaluates whether executing ML tasks with the given data are resource-intensive or not. If the module determines that performing ML tasks at the edge is advantageous, the tasks are carried out there. By leveraging AI-driven decision-making, that framework seeks to optimize data handling and processing between the edge and the cloud, contributing to improved DR and communication efficiency.

Indeed, several studies have leveraged the capabilities of AI to address DR challenges in the context of IoT. The work [22] focuses on DR at the edge using AI techniques. It presents a framework that employs feature selection and extraction mechanisms to reduce the amount of data generated by sensors. The authors of [23] propose a solution that combines cloud and edge computing for data analytics. They implement a DR strategy on the edge, followed by data transmission to the cloud for further processing using Autoencoders (AE). That approach achieves around 50% DR with minimal accuracy impact. The authors of [24] present a framework that employs AE deployed at the edge to perform DC on raw data. The goal is to transform high-dimensional data into compact forms, reducing data volume without significant loss of information. In all these works, different strategies are employed to achieve efficient DC.

A Systematic Literature Mapping (SLM) was conducted in [12], addressing DR solutions at the edge of IoT systems. That article aimed to understand the intricacies and distinctive attributes of implementing DR within resource-constrained devices deployed at the edge of the network, highlighting the increasing importance of reducing the volume of data generated at the edge-sensing layer. In general, it suggests a growing recognition of the benefits and significance of managing data volume directly at the point of data generation within IoT architectures.

The authors of [25] introduced deep learning models to the fog with the goal of enhancing video transmission while simultaneously minimizing data use through the extraction of video features. Their resolution technique effectively distinguished particular and potentially significant regions with high precision, whereas lower-quality data were utilized for areas of lesser importance. The authors used the DeepLabV3+ model to extract the features of the video, getting 45% higher average speed while comparing with Resnet-101. After identifying the salient zones of the video, they are transmitted to the compression phase. The authors argue that a reduction in 71.02% of a system delay is reached. Initially, the IoT sensor cameras capture images and videos sending than to a server (e.g., Personal Computer (PC)), which is categorized as a fog node. There, they perform feature extraction functions while discarding redundant information to save space and reduce data volume. They also introduced an energy function to support such demands. Finally, before transmitting to the data center which is located on the cloud, the authors implemented a compression approach to the data, the High-Efficiency Video Coding (HEVC), which is a lossy compression codec algorithm. Therefore, the data transmitted over the network was reduced, resulting in up to 71% system reduction delay.

The framework model introduced at the edge, based on auto-encoders [24], is focused on the application of the DC DR technique to the raw data received from sensors. By utilizing this approach, it guarantees that the initial data can undergo preprocessing prior to being transmitted to the cloud. This results in the transformation of high-dimensional data into more condensed data. The autoencoder’s encoder component is concentrated on DR and runs on the edge, whereas its decoder component is focused on image reconstruction and classification and operates on the cloud. In their work, the authors put forth three potential solutions: Edge, Cloud, and Edge+Cloud. After the image data from the IoT camera is transferred to the edge buffer, the edge approach compresses the data before transmission to the cloud. By utilizing this approach, the final performance of the system is notably enhanced as the network cost of the edge–cloud connection is reduced. The approach that is known as Edge+Cloud, or the hybrid approach, ultimately acknowledges that the buffer is completely full. The data that are received in this instance are transmitted unadulterated and without any form of compression to the cloud. In the author’s solution, the experiment was simulated and evaluated using the SimPy [26] discrete-event simulator. The author’s evaluation also presented different buffer size configurations that impacted the total bandwidth consumed results.

Researchers in [27] proposed a compression scheme that is introduced at the edge to reduce the signal size before its transmission over the network. They combined a Convolutional Denoising Autoencoder (CDAE) and LSTM focusing on signal compressing and denoising to reduce the transmission cost. The author’s solution is built considering three fundamental requisites, which are (i) compression efficiency, (ii) reconstruction quality, and (iii) energy consumption. As the author’s solution is constructed considering the constrained memory and energy resources devices, the reduction in the energy consumption is primordial when implemented in IoT scenarios. The author’s proposed solution is composed of the introduction of an LSTM network at the end of the encoder section of the CDAE for reconstructing the ECG signal. It also introduces a method that denoises the sensing ECG data from its noise version with better accuracy than previous research.

In [28], the authors proposed a feature selection DR approach for ML at the fog layer focusing not only on reducing the data volume transmitted to the cloud but also on optimizing the model learning capabilities. The attribute analysis techniques used to reach the reduction used by the authors were Gain Ratio Attribute Evaluation, Info Gain Evaluation, CF Subset Evaluation, and Principal Component Analysis. The configuration parameters for the aforementioned algorithms were 0.5 for feature parameters, and only the Principal Component Analysis technique was configured with a variance of 95% in all the other cases. The metrics used by the researchers to analyze the results are composed of Precision and Recall in the form of an F-measure. The author’s solution reduced 50% of the total features, providing results where the obtained set performance was increased or similar to the full set of features while the ML model performance was not significantly affected by the reduced features.

The authors of [29] proposed a compression and learning framework aiming to improve the coding rate and accuracy performance classification on images. They used a specific Variational Auto Encoder (VAE) type for compression and classification. Their solution uses latent vectors without reconstruction of the image to classify the images in the cloud. Their solution is motivated by the fact that the classification accuracy of an image compressed by conventional codecs suffers at high compression ratios and also demonstrates an increase in inference speed and accuracy degradation. It is composed of an Autoencoder, a Probability Estimator (PE), and a classifier (CL). The ResNet network was split into two parts, one as the encoder and the other as the classifier. The dataset used for experimentation comprised JPEG images in which compression was reached through different quality values.

The study [30] proposed an image compression framework built specifically for running in resource-constrained IoT devices. The solution uses a DNN that avoids inefficient image transmissions being projected to low-memory and processing IoT devices. The project of constructing a DNN or any other AI module aiming for its implementation in accelerators and microcontrollers is challenging once there are many memory, storage, computing, and energy constraints. Therefore, considering such factors, the authors used the AutoML technique when constructing the DNN, which was constructed using the Network Architecture Search (NAS) method. The results of its framework presented results up to 3 and 4 times in image compression, whereas the energy efficiency traffic performance was up to 2.5 times.

Overall, previous works have presented promising contributions to the use of AI-based techniques for data reduction, but some important gaps remain. For instance, Table 1 summarizes the presented works according to their proposed DR technique, employed AI algorithms, and the logical layer where such proposals were implemented for both elements. We noticed that most data reduction solutions discussed herein use data compression as a DR technique and AE was the AI solution. Actually, our approach innovates by performing context-aware optimizations for better performance, supporting IoT applications in different areas. Moreover, our solution allows several DR algorithms to be introduced into the framework considering all the techniques (e.g., Data Compression, Data Prediction, Data Sampling, etc.) presented in Table 1. This factor allows heterogeneous data to be treated and processed in order to reduce their volume in IoT environments. Furthermore, our work proposes an AI-based DR framework that predicts values for the metrics presented in Section 3.6, filling the column that is missing in Table 1 as DR solution.

## 3. Proposed Framework

Having a unique DR solution capable of handling all data types effectively in an IoT environment seems to be an infeasible option. It happens because IoT systems, and notably smart city applications, have a big heterogeneity of data and thus need particular and individualized treatments. Conversely, offering a solution that exclusively caters to a single scenario and context, resulting in a specific data type, might also be unsuitable. Therefore, in this article, we propose a context-based solution that switches among the DR algorithms depending on (i) the predetermined context relevance of each data chunk, (ii) the RR projection of each algorithm, and (iii) the portion of data distortion that is directly related to its context relevance. For that, a comprehensive multi-tier framework is proposed, allowing practical exploitation of the proposed solution.

The intended characteristics of the proposed approach are highlighted as follows:It considers heterogeneous workload data types. The solution is not fixed in one data type; instead, it can handle any data towards reducing its volume without compromising quality;It considers many DR algorithms as options to the model selection. Depending on the used data, there are classes of reduction algorithms that perform better than others. Hence, our solution is projected to select the algorithm that presents the best parameters (e.g., reduction ratio and distortion), considering the context of reduction quality that a data point is allowed to have;It uses multiple architectural layers in its implementation, being focused on computing closer to the edge of the network. Doing so, it would reduce bottlenecks relating to the transmission, storage, and retrieval of data;It considers the context and relevance of each data producer. As each producer is early classified, the solution considers the importance of the data producer to select the reduction algorithm that is the best fit;It is projected for implementation in multiple scenarios, not being restricted to one domain since it is a generic framework;It includes an intelligent module to predict reduction performance and decrease data distortion, increasing the processing performance. The selected model is responsible to predict the metrics on the edge to avoid unnecessary computing, improving many applications performance directly on the sensor or edge layers.

The next subsections present the proposed framework explaining the main modules and their engineering traits and details, as well as the expected operation flow.

### 3.1. Conceptual Operation

The proposed multi-tier framework consists of three layers: the Sensor Layer, the Edge and Fog Layer, and the Cloud Layer. It is worth noting that, in this work, the developmental research centers around the edge layer, as DR is expected on it. Nevertheless, the entire suggested data flow demonstrates a scenario where data are generated at the sensor layer and subsequently utilized by the ultimate application at the cloud layer. The emphasis lies on edge processing since managing diverse sensor data in proximity to its origin could potentially enhance the performance of the IoT architecture. Actually, sending the raw data from sensors to the cloud without preprocessing could lead to network bottlenecks, heightened energy usage, device deterioration, and a range of other detriments that reduce system performance.

Considering that our emphasis is on minimizing data at its source and maintaining its quality, it is important to have this functionality as close as possible to the data production, which is conceptually encompassed by the edge layer. Figure 1 illustrates the overall data flow of the proposed framework.

Initially, the sensor acquires environmental data by being strategically positioned across distributed environments. Prior to transmitting the recorded data, it undergoes initial compression using traditional algorithms like Huffman Coding [31], Run-length Encoding [32], Arithmetic Coding [33], Lempel–Ziv–Welch, and Shannon–Fano Coding. Based on the attributes of the raw data, a broad variety of compression algorithms could be employed during this data-capturing step.

In such scenarios, multimedia data necessitate special considerations. Applying traditional codecs to compress multimedia data reduces the data’s quality, thereby impacting its performance when utilized by AI models, as evidenced by studies such as [30,34]. To address this issue, researchers are turning to deep learning approaches such as Convolutional Neural Networks (CNN) and Deep Neural Networks (DNN) for handling this type of data. As an example, in their work, the authors of [30,34] employed Deep Neural Networks (DNN) to handle multimedia data. Opting to compress image data using traditional image compression codecs before transmission is not advisable, as this approach leads to the removal of crucial image structures that DNN models rely on. Furthermore, employing such codecs to compress image data can result in a significant increase in inference error bounds.

Upon reaching the edge, the data generated by the sensor undergoes manipulation through a fundamental sequence of five steps. Data decompression stands as the initial manipulation that the data undergoes. At this stage, the data are usually ambiguous, unclean, and unsuitable for storage and analysis. Every generated data point must be temporarily stored in a data pool, facilitating downstream processes to systematically pass it on to the subsequent stages of processing. Subsequent to this data-capturing phase, the data-handling phase begins. The primary objective of this step is to transform the available data into a succinct and efficient perspective.

The captured data undergoes data clustering and is then fed into an unsupervised machine learning model. At this point, DR methods, like DC, can exhibit improved efficiency when dealing with clustered data, as discussed in [35]. The decision between choosing a supervised or an unsupervised ML model was also influenced by the clustering module as a determining factor. Furthermore, addressing data heterogeneity entails managing data patterns and unforeseeable characteristics that would be explored by the ML model.

As this proposal needs a trained ML model to be deployed in the edge devices, it is worth noting that the model is initially trained on the cloud or a different server with ample processing power compared to the edge. It is common practice to use a robust server rather than edge devices to train AI models due to its processing power, although development trends in training on the edge are also being developed. The proposed framework encompasses a set of DR techniques and algorithms designed to reduce the dataset in terms of volume, complemented by a ML model. The ML model aims to decrease data volume using various DR algorithms by initially adjusting three parameters: (i) context relevance; (ii) Distributed Function Performance (DFP); and (iii) the Reduction Ratio (RR) of each segmented data chunk.

As discussed before, having a unique DR solution capable of handling all data types is not efficient. Our proposed context-based solution switches among the DR algorithms depending on predetermined context relevance, the RR projection of each algorithm, and the portion of data distortion that is directly related to its context relevance. Based on the specified values, an ML model should determine the suitable DR technique that offers superior overall performance for DR. In our solution, the context relevance of each data producer needs to be established prior to gathering data from sensors. It will indicate the acceptable distortion variance the data can acquire after a reduction algorithm is applied to the data chunk.

It is important to remark that data chunks that were classified with higher relevance are not allowed to be reduced by lossy DR methods. This could compromise the quality of data, which holds significance within a specific domain. Hence, this particular category of data should be reduced using lossless DR algorithms, guaranteeing the potential for original data reconstruction when needed. On the other hand, data chunks characterized by moderate and low relevance can undergo reduction through lossy DR algorithms. The rule governing this decision relies on the configuration parameters that are predetermined before the ML prediction. The initial configurations are outlined in Table 2, serving as the basis for subsequent evaluations.

### 3.2. Data Acquisition and Reduction

All data producers need to be configured based on one of the provided relevance levels (e.g., Very Low, Low, Moderate, High, and Very High). As the relevance level increases, the DFP value decreases, indicating a tolerance distortion for the data chunk. Table 2 illustrates the correlations between metrics for the predicted data producers. On the one hand, it is expected that the DFP classification consists of predetermined values that should be adapted based on the model’s performance enhancement. These percentages are measured within a range spanning from 0% to 100%. The singular predefined restriction is that the DFP for the Very High relevance level must be set to 0%; thus, “x4” in Table 2 must be 0%. On the other hand, RR metrics provide information about the extent to which the volume of data might be reduced if a particular DR algorithm is applied to such a data chunk. Once the model’s framework computes and predicts these metrics, it will choose a suitable DR algorithm based on the comprehensive performance assessment. This selected algorithm will be employed to reduce the data before transmitting it to the cloud.

Data loss distortion x% and x1% mean the acceptable distortion a data chunk can perform; in such a case, it relates to the Very low and Low relevance, respectively. In smart cities, nodes exhibit varying contexts with relevance spanning from Low to Very High due to the heterogeneity of the produced data. Some contexts involve sensors reading data with minimal variation (e.g., temperature), which do not need to be stored continuously. Therefore, these readings can be categorized as very low or low-relevance data producers, permitting some data loss in the reduction phase. For all relevance levels except very high, both lossy and lossless DR algorithms can be used. However, in scenarios with extremely high accuracy requirements, the use of lossy DR techniques might not be permissible.

The DR solution is implemented after the models’ consideration towards reducing the data size of the clustered chunk data. Many DR techniques (e.g., data filtering, DC, data sampling, etc.) could be applied to reduce data volume by the intelligent model depending on the data characteristics. This is depicted in Figure 1 through a series of DR blocks ranging from 1 to “ n”, signifying the existence of “ n” solutions for reducing data volume.

DR algorithms are categorized as either “ lossy” or “ lossless”. In the former, data loss occurs during the reduction process, whereas, in the latter, the original data meaning remains intact. In environments such as hospitals, airports, etc., data loss is impermissible; thus, for those classes of environments, the lossy algorithms must be avoided.

Furthermore, it appears beneficial for enhancing the model to store local information about the DR algorithms used for the data chunk reduction. Hence, a local data storage system responsible for storing these reduction decisions is reserved to be implemented at the edge. The reduced data are then transmitted to the cloud layer.

High-level information data are stored in the cloud. At this stage, raw data are not stored in the cloud; instead, the information is preprocessed, categorized, cleaned, and adapted for storage. Lastly, the application utilizes these data for its intended purpose. The cloud-stored data can also be leveraged for retraining purposes. Cloud storage typically boasts more robust capabilities than edge storage, making this module crucial for fulfilling such requirements. Within this context, retraining holds the potential to enhance the model’s performance.

### 3.3. Sensor Flowchart

Sensors are commonly distributed throughout environments to capture data regarding a relevant occurrence. The sensors flowchart of the proposed framework is depicted in Figure 2.

A sleep–wake cycling mechanism that leads to increased energy consumption in comparison to an “all-times-on” system is usually employed in this kind of system. As soon as the data are read, the sensor enters an active state, measuring a new value, X, which is then compared to the last recorded data, Y. Considering that X and Y are data readings, when X equals Y, it indicates that the new X data does not provide extra information compared to the previous one. This situation prompts a decision to abstain from transmitting this X data, thereby conserving resources. Conversely, when X and Y differ from each other, the validity of the X data are assessed to eliminate potential outliers or corrupted data before transmission. In instances where the X data are confirmed as valid, it undergoes compression as ’X and is subsequently dispatched to the edge.

### 3.4. Edge Flowchart

The flowchart in Figure 3 outlines most of the procedures involved in edge data processing. As soon as the data reach the edge, the compressed X data undergo decompression, reverting to their original X form, and are then archived within a data pool.

The following are the main reasons for the created data pool.

It must have the capacity to retain all generated sensor data without any loss. In IoT scenarios, there is a prodigious influx of data, and, for certain data types like multimedia, their size can increment exponentially. Moreover, certain contexts, such as health and airport data, are crucial, with data loss being unacceptable and inadmissible;The data pool needs to be designed to store heterogeneous data captured from diverse environments. Sensors producing heterogeneous data exhibit distinct data traits that necessitate preservation and proper handling through their respective modules. Hence, if any module issue occurs, the data should be securely stored and sustained within the data buffer;Finally, it should facilitate synchronous data provision to the clustering block, effectively managing transmission throughput and local storage capacity to avert overwhelming situations. Depending on the data transmission configuration, data might need to be organized into transmission blocks, and having a specific local for that can significantly enhance performance.

Following the temporary retention of the generated data chunks, they are directed to the clustering phase. In this stage, the data are structured, segmented, and categorized for utilization as input by an unsupervised ML model. Typically, unsupervised ML models exhibit good performance when applied to clustered data, as opposed to unorganized data. Hence, the clustering of heterogeneous data holds significant importance within this solution.

Upon completion of the clustering process, the model undertakes the task of assessing crucial DR attributes of the clustered dataset, considering various DR methodologies. Using the data chunk as input, the model provides estimations for both attributes of each DR algorithm (e.g., DFP and RR). The former provides a distortion rate for each DR algorithm, whereas the latter quantifies the potential reduction in data volume for each technique. DFP serves as the guiding metric for selecting the appropriate DR algorithm, and its determination is directly tied to the data’s relevance, as outlined in the preceding section. This metric operates based on the subsequent set of rules:Each data chunk is assigned a predefined relevance value that falls into one of five classes: Very Low, Low, Moderate, High, and Very High;The ML process will make predictions for the DFP and RR parameters of all “*n*” DR algorithms;Based on the projected *Distortion* value, an appropriate DR algorithm—either Lossy or Lossless—will be selected, ensuring that the acceptable threshold of distorted data is upheld during the DR process;If a lossy DR algorithm is available to the data chunk, and it respects the pre-established distortion, it will be employed for DR; a lossless DR algorithm will be utilized if not;Historical ML projections can potentially be stored in the local edge storage and later used for retraining purposes.

Following the model’s predictions, a suitable DR algorithm is chosen to effectuate DR, respecting both distortion and relevance parameters. The projected reduction outcomes for all data chunks are archived within the local edge storage. In general, this data storage feature offers intriguing possibilities: it can be used for retraining the projected model or even reconstructing the original data. Therefore, it is important to store such projections within this layer. Lastly, the data are transmitted to the cloud.

### 3.5. Cloud Flowchart

Figure 4 depicts the cloud flowchart of our framework. Within the cloud layer, a reduced dataset is received, distinct from the originally generated raw data. At this step, the data have been clustered, cleansed, and projected to provide reduction information for the model. If the data had been previously stored, it necessitates removal; otherwise, they may be stored within the cloud repository.

Concerning the model, each instance of information storage in the high-level database warrants an assessment of the necessity for model retraining. Upon confirmation, model retraining takes place, and the outcome is relayed to the edge for performance enhancements. Lastly, an application can draw upon the database information for analytical or other tasks.

### 3.6. Framework Metrics Prediction

The study primarily focuses on the implementation of DR algorithms on the edge of the network that focus on reducing transmission overhead. This reduction occurs considering that the model’s main function is to predict some metrics, as discussed below. Considering the quantity of existing DR algorithms in the literature and the heterogeneity of the data, this is a challenging task. Therefore, the metrics discussed below are important for the model prediction.

Directing attention towards the edge layer, after decompressing all the received data from sensors, it is aggregated within a pool and subsequently employed as input for the clustering process. The initial phase of experimentation is tied to the clustering procedure. However, the principal evaluation of this layer is associated with the assessment of the RR and DFP of the resulting data. Thus, the analysis should be directed towards the reduction of these evaluation metrics.

According to [27], the evaluation of any compression technique can be carried out by taking into account various metrics: Quality Score (QS), Compression Ratio (CR), Root Mean Square Error (RMSE), Percentage Root Mean Square Difference (PRD), and Signal to Noise Ratio (SNR). Considering we are experimenting with compression algorithms, we selected the CR as RR metric to be used in this study. In the literature, various DFP measurement variations are available. In this framework, in addition to the PRD, we employ the two most prevalent *difference distortion measures* metrics, namely, (i) the Mean Squared Error (MSE) and (ii) the Mean Absolute Error (MAE).

In fact, the *CR* metric serves as the cornerstone for evaluating compression efficiency, being calculated by dividing the size of the original data 
$S_{o}$
 by the size of the compressed data 
$S_{c}$
, as expressed in Equation (Equation 1).

(1)
$$CR=\frac{S_{o}}{S_{c}}$$


The measurement of error between the original data and the signal that has been reconstructed is known as *PRD*. This value, presented in Equation (Equation 2), serves as an indicator of the quality of the reconstructed signal. Furthermore, it offers a quantification of the error existing between the original data and the reconstructed signal. This measurement consequently provides insights into the quality of the reconstructed signal.

(2)
$$PRD=\sqrt[{2}]{\frac{\sum_{i=1}^{N}{\left(x_{i}-y_{i}\right)}^{2}}{\sum_{i=1}^{N}{\left(x_{i}\right)}^{2}}}\times 100$$


Equation (Equation 2) outlines the computation of PRD. Here, 
$x_{i}$
 represents the original signal values, and 
$y_{i}$
 represents the corresponding values in the reconstructed signal. The summation is carried out over *N* data points, and the result is multiplied by 100.

Lastly, the *QS* metric, presented in Equation (Equation 3), provides an expression of compression efficiency by dividing the *CR* by the *PRD*.

(3)
$$QS=\frac{CR}{PRD}$$


## 4. Evaluation Methodology

This section introduces the experimentation process and results that were developed through this research. Actually, the optimal performance of the parameters outlined in Section 3 is the main focus of the experimental evaluation in this work. In such a case, the data producer relevance requirements are satisfied by the lowest DFP and highest RR. To achieve that, the relevance of the data producer is merged with an acceptable data distortion from the dataset towards reduction. Given that the framework solution comprises three layers, an experimental projection would be provided for each of them.

A variety of metrics can be used to evaluate our proposed framework. Given that the primary emphasis during this initial step is on the utilization of exclusively lossless DR algorithms to achieve the reduction of time-series data, the assessment in this stage exclusively utilizes the CR metric. The next subsections further discuss the defined evaluation methodology and parameters.

### 4.1. Experimental Environment

In pursuit of a more adaptable platform, the experimental environment for evaluating this framework encompasses the simulation within a Virtual Machine (VM). This VM is equipped to simulate diverse sensor devices, as exemplified in [14], and facilitates virtual interactions between sensors through a local network. For evaluating the sensing layer, we programmed the compression algorithms presented in the next section towards identifying better reducibility. The objective of this setup is to simulate the heterogeneous sensors dispersed across the network, thereby validating the efficacy of the proposed framework.

From an experimental perspective, data transmissions should be managed to accommodate two potential scenarios, which will be considered as the evaluation reference in this work:Scenario 1—Grouped Transmissions: sensor data transmissions occur by bundling a specific batch of data together before transmission;Scenario 2—Individual Transmissions: each sensor transmits data in real-time as soon as they are read, as depicted by the sensor flowchart presented in Figure 2.

### 4.2. Evaluation Algorithms

The aim of this section is to identify the most effective compression algorithm to be implemented within the sensor devices. Nonetheless, given the heterogeneous nature of the selected scenario, we focus on sensor devices that generate text data. This data format is chosen as it aligns with the predominant data type produced by a majority of IoT technology sensors, specifically time-series data. According to [36], a substantial proportion of data generated by IoT devices is in the form of time-series data. Time-series data are characterized by an additional dimension, time, in contrast to tabular data, including features and instances. This enables time-series data to incorporate more instances of data points compared to tabular data.

In this experiment, five lossless compression algorithms tailored for compressing sensor time-series data were employed:Arithmetic Coding [33];Huffman Coding [31];Lempel–Ziv–Welch [37];Run-length Encoding [32];Shannon–Fano Coding [38].

This particular phase of the framework intentionally excludes considerations of lossy DR methods. The rationale behind this exclusion is two-fold. On the one hand, applying lossy DR algorithms to compress text data are not conventionally sound. Lossy solutions entail a trade-off between data quality and data volume reduction. In the case of text data, a reduction in quality could lead to modifications in the message conveyed by the file, potentially rendering it unintelligible. On the other hand, lossy DR techniques often require extensive processing and analysis to identify features and patterns for removal, resulting in a reduction of data size. However, sensors are constrained devices that can face limitations when burdened with resource-intensive tasks. Thus, in this phase, the focus remains solely on employing lossless compression algorithms.

### 4.3. Considered Dataset

To emulate the sensor data readings on the sensor layer, the dataset outlined in [39] has been chosen for utilization in the experiments. Although our framework is designed to accommodate heterogeneous data transmissions, at this phase, text-based time-series data have been chosen to illustrate the analysis of compression algorithms.

The chosen real-world time-series dataset encompasses temperature and humidity monitoring readings. The data spans a period of 4.5 months, captured by IoT devices equipped with appropriate sensors, and averages 19,735 readings over 10 min intervals per wireless transmission. To create this dataset, the authors in [39] employed 9 Arduino boards, each paired with an XBee radio transmitter (IEEE 802.15.4). Moreover, those devices were strategically placed to collect data from the environment, which were then transmitted to another XBee radio configured as a network coordinator (gateway) within a house. All Arduino boards were battery-powered. More details about the configuration of the Arduino boards and their actual deployment can be verified in [39].

Finally, this dataset is organized into two distinct configurations, which are discussed in the next section. Although the dataset is more related to a smart home perspective, we believe it represents a small instance of a smart city when concerning the existing challenges and data heterogeneity.

## 5. Experiments and Results

After the careful definition of the proposed approach and detailed explanation about the evaluation methodology, a series of experiments was conducted aimed at validating our work. First, the evaluation algorithms were considered for the defined dataset, demonstrating how efficient DR could be achieved. Then, discussions about the practical exploitation of this framework in IoT scenarios are presented, especially focused on smart city applications. We believe that the discussed results are important contributions toward the practical adoption of the proposed approach.

### 5.1. Numerical Results

As an initial evaluation step, we considered the exploitation of the proposed framework taking a smart city scenario as reference. Smart cities are distinguished by the existence of numerous data nodes that generate data at a substantial rate, leveraging communication resources to address intricate daily tasks. The data generated within this context can be generated from diverse source nodes, such as individuals, vehicles, residences, industries, hospitals, and much more, spanning different locations and operating concurrently. Each of these sources contributes to an inherent significance to the extracted data, thereby enhancing the contextual understanding of urban environments. With all that said, we believe this is a potential IoT scenario for improvement when adopting an efficient DR approach.

As aforementioned, we have analyzed the five evaluation algorithms within the context of two well-defined data configurations, according to the defined parameters for the construction of the employed datasets. In short, the configurations are:Configuration 1: nine sensor readings are stored per line in the CSV file, resulting in a total of 19,735 readings, each approximately 549 bytes in size;Configuration 2: each CSV line contains the readings from a unique board. This configuration yields a total of 177,615 data streams, each approximately 241 bytes in size.

The data specifications for both Configuration 1 and 2 are exemplified in Table 3.

For the two defined configurations, we aim to determine which algorithm presents the highest CR, as this is the chosen evaluation metric. The results are presented in Table 4.

The experiment was initiated by compressing the original data from both configurations separately. For Configuration 1, the original data file size amounts to 11,637,491 bytes. In this case, the compression process yielded interesting outcomes. The Huffman, LZW, and Shannon–Fano algorithms demonstrated similarities in reducing the original data to represent 45.50%, 50.92%, and 54.05% of the original file, respectively. Such algorithms presented RRs of 54.5%, 49.08%, and 45.95% and CRs of 2.19, 1.96, and 1.85, respectively. These outcomes reflect substantial compression efficacy, potentially contributing to time and energy savings on resource-limited devices.

The Arithmetic Coding algorithm exhibited significant reduction performance, reducing the data to 25.71% of the original size, with an RR of 74.29% and a CR of 3.89. However, this enhancement in compression came at the expense of increased processing time. While other algorithms were executed within minutes, Arithmetic Coding’s processing times extended into hours. It is important to note that, as the primary goal of this experiment was to evaluate volume reduction, the latency introduced during this preprocessing was not considered. Lastly, the Run-length Encoding (RLE) algorithm yielded the poorest compression performance, increasing the data size by 3.94% and presenting a CR of 0.96.

Analogous to the initial experimentation, results for Configuration 2 are also relevant, where 177,615 time-series data streams, collectively totaling 52,218,939 bytes, underwent compression. Huffman (53.45%), Arithmetic Coding (55.40%), Shannon–Fano (66.48%), and LZW (67.79%) were the values for the original data representation, in addition to their CRs equal to 1.87%, 1.80%, 1.50%, and 1.47%, respectively. However, the computational time pattern observed in Configuration 1 persisted here. While Arithmetic Coding required over a day to compress all stream files, the other algorithms were accomplished within minutes. In contrast to Configuration 1, in Configuration 2, the Huffman algorithm exhibited the best compression performance. Lastly, the Run-length Encoding (RLE) algorithm displayed sub-optimal performance, increasing the data size to 19.02% with a CR of 0.84.

The objective of these experiments was to determine which compression algorithm yielded the best CR for time-series data. The results revealed that both the Huffman and Arithmetic Coding algorithms were top performers in terms of size reduction for both configurations. In Configuration 1, the Arithmetic Coding algorithm presented superior performance with a CR of 3.89, followed closely by the Huffman algorithm with a CR of 2.19. In Configuration 2, which more closely aligns with our model, the Huffman algorithm showcased the best CR at 1.87, with the Arithmetic Coding algorithm following closely at a CR of 1.80. Given that Configuration 2 is more representative of our model, the Huffman lossless DR algorithm was selected for implementation within the sensor devices of our proposed solution.

As a last remark, while the processing time of compression algorithms was not directly evaluated in this experiment, it remains crucial to minimize it in order to mitigate latency during the transmission of sensor data. Such concern will be further evaluated in future works.

### 5.2. Practical Applications and Envisioned Scenarios

In the rapidly evolving landscape of smart cities, the efficiency and sustainability of IoT-based applications play a pivotal role in shaping urban development. Harnessing the power of diverse and heterogeneous data sources, these applications provide valuable insights to enhance city services, optimize resource utilization, and improve the overall quality of urban life [5,6]. In this context, the advent of an innovative framework capable of reducing the total amount of data to be transmitted over the network holds immense promise for the future of smart city technologies.

As the core idea behind our approach lies in its ability to significantly diminish the amount of data to be transmitted over the network, it not only reduces the demand for transmission bandwidth but also incurs energy saving on different elements of the considered systems, which are highly desired goals for smart cities in general [40,41]. As smart cities continue to proliferate with a multitude of interconnected devices and sensors, energy efficiency becomes a critical consideration. Therefore, by alleviating the burden on network infrastructure and the associated energy-intensive processes, our approach emerges as a cornerstone for sustainable and eco-friendly smart city ecosystems.

Other major issues of the proposed framework lie in its adaptability to the inherently heterogeneous nature of IoT-based applications, which is an expected reality for smart cities. With a myriad of devices, sensors, and data streams contributing to the city’s operational intelligence, managing this diversity efficiently is key to extracting meaningful insights. The framework seamlessly operates for diverse data sources, optimizing the flow of information closer to the data sources and ensuring that only reduced/compressed data are transmitted across the network.

The following smart city applications are some examples that might directly benefit from the proposed approach:Smart mobility: traffic monitoring systems in smart cities rely on real-time data from various sources, including traffic cameras, sensors, and GPS devices. The proposed approach can significantly reduce the amount of data transmitted for traffic analysis and optimization, leading to faster decision-making for traffic management systems;Environmental monitoring: smart cities may deploy numerous sensors to monitor air quality, pollution levels, and other environmental parameters. This way, the framework can filter environmental data at the source, allowing for more efficient utilization of resources;Public safety: surveillance cameras, IoT devices, and other sensors contribute to public safety and emergency response systems. The proposed approach can streamline the processing of surveillance data at the edge, enabling quicker detection and response to potential security threats. This not only enhances public safety but also conserves bandwidth and reduces energy demands;Sustainable waste disposal: these systems use sensors to monitor waste levels in bins and optimize collection routes. Our proposal can minimize the data sent from waste sensors, allowing for efficient and timely waste collection without unnecessary data transmission.

This way, we believe that the proposed framework may represent a paradigm shift in the development of IoT-based applications for smart cities, acting as an important reference for new implementations. As cities continue to evolve into intelligent, connected ecosystems, the amount of data to be transmitted, processed, and stored daily may be huge. Then, our approach stands out as a viable solution for more efficient and scalable management of data sensing in smart city applications.

## 6. Conclusions

Heterogeneous data are generated by a diverse range of sensor devices across various contextual environments. These sensors are strategically situated to capture local digital/analog data, resulting in the generation of yottabytes of heterogeneous IoT sensor data in smart city applications. Managing heterogeneous data, however, is not a straightforward endeavor. However, prudent organization of generated data can alleviate access bottlenecks and enhance data utilization.

Implementing data reduction methods to eliminate non-useful and duplicate data holds several computational advantages, including conserving network bandwidth, reducing energy consumption, optimizing storage utilization, and minimizing traffic costs. Given the diverse possibilities and techniques inherent in implementing DR methods within an edge–fog–cloud architecture, this research work advocates an edge AI-based DR framework. Our proposed framework comprises three main modules (sensor, edge, and cloud), each with distinct roles, being a valuable addition to the development of IoT-based applications. Five compression algorithms were evaluated under two data production configurations. The Huffman compression algorithm was chosen to be employed in the first stage of the framework due to its strong performance. An unsupervised ML model was also employed to reduce data volume while preserving data quality through the analysis of RR and distortion metrics. After experiments, an initial evaluation stage satisfactorily indicated the applicability of the proposed approach in a real scenario.

In future works, the intended evaluations will also be composed of latency analysis, as this directly impacts the feasibility of the proposed solution. Additionally, the accuracy of the reduced data will be integrated into the cloud, enhancing the model’s performance by effectively reducing data in a concise manner. Finally, larger datasets will be considered, potentially taking data from large smart city systems, further validating the proposed approach.

## Figures and Tables

**Figure 1 sensors-24-00358-f001:**
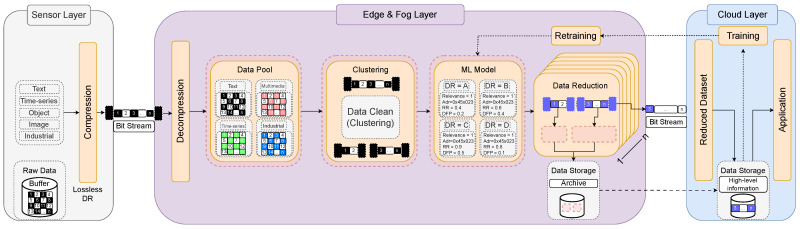
Conceptual organization of the proposed multi-tier framework.

**Figure 2 sensors-24-00358-f002:**
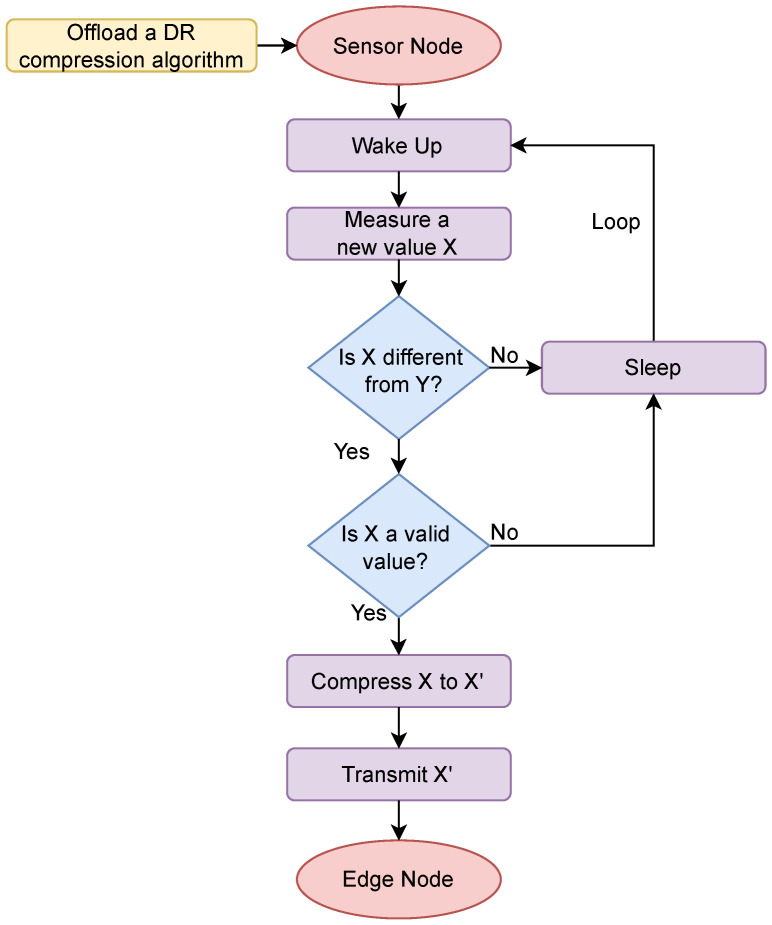
Sensor data transmission flowchart.

**Figure 3 sensors-24-00358-f003:**
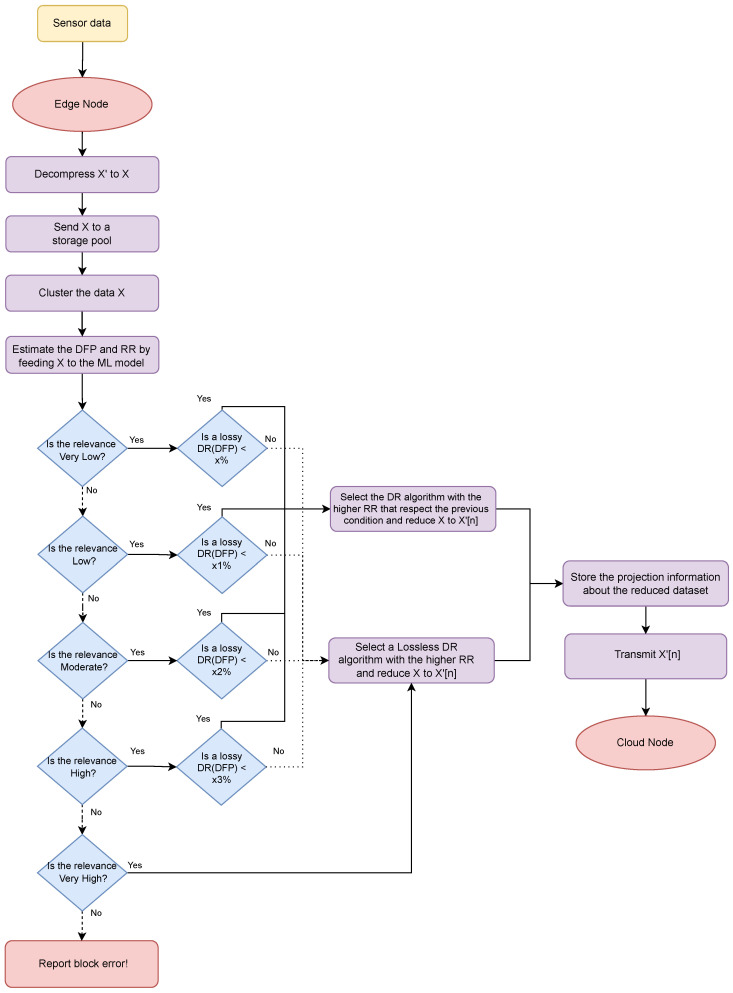
Edge data transmission flowchart.

**Figure 4 sensors-24-00358-f004:**
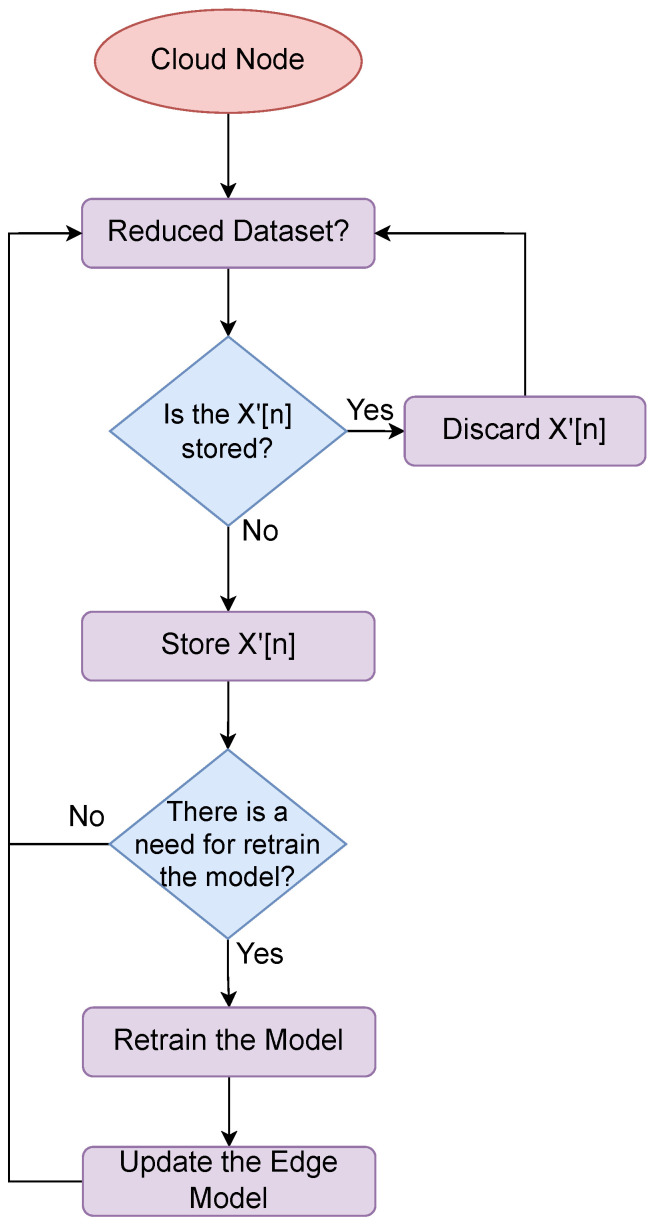
Cloud data transmission flowchart.

**Table 1 sensors-24-00358-t001:** Relation between DR and AI techniques in previous works, highlighting the conceptual layer where each approach is applied.

Work	Year																			
		DR Layer			DR Technique						AI Layer				AI Technique					
		Sensing	Edge	Fog	Data Compression	Data Prediction	Data Sampling	Feature Selection	Feature Extraction	Data Filtering	Sensing	Edge	Fog	Cloud	Non-DL-ML	DNN	CNN	RNN	AE	RL
[21]	2019		✓							✓		✓		✓	✓					✓
[25]	2019			✓	✓				✓								✓			
[23]	2019		✓					✓				✓		✓	✓				✓	
[24]	2020		✓		✓							✓		✓					✓	
[30]	2020	✓			✓						✓					✓				
[22]	2021		✓		✓		✓			✓		✓		✓	✓					
[27]	2021		✓		✓							✓					✓	✓	✓	
[28]	2021			✓				✓					✓		✓					
[29]	2022		✓		✓							✓			✓				✓	

**Table 2 sensors-24-00358-t002:** Distortion Function Performance (DFP) and Reduction Ratio (RR) Classification.

Relevance	Very Low	Low	Moderate	High	Very High
DFP (%)	x	x1	x2	x3	x4
RR	x^′^	x1^′^	x2^′^	x3^′^	x4^′^

**Table 3 sensors-24-00358-t003:** Data structure presenting some samples with rounded values.

Header	*date, Appliances, lights, T1, RH_1, T2, RH_2, T3, RH_3, T4, RH_4, T5, RH_5, T6, RH_6, T7, RH_7, T8, RH_8, T9, RH_9, T_out, Press_mm_hg, RH_out, Windspeed, Visibility, Tdewpoint, rv1, rv2*
Configuration 1	2016-01-11-17:00:00, 60, 30, 19.890, 47.597, 19.199, 44.789, 19.789, 44.729, 19.000, 45.567, 17.167, 55.200, 7.027, 84.257, 17.199, 41.627, 18.199, 48.899, 17.033, 45.530, 6.599, 733.500, 92.000, 7.000, 63.000, 5.299, 13.275, 13.275
Configuration 2	2016-01-17-13:50:00, 410, 0, 22.000, 38.229, 2.983, 765.417, 64.333, 2.000, 40.000, —3.250, 21.749, 21.749

**Table 4 sensors-24-00358-t004:** Achieved numerical results for both scenarios.

Algorithm	Configuration	Original	Compressed	Compressed	CR
		Data (B)	Data (B)	File Avg. (B)	
AC	1	11,637,491	2,991,552	151.58	3.89
Huffman	1	11,637,491	5,295,042	268.30	2.19
LZW	1	11,637,491	5,925,544	300.25	1.96
RLE	1	11,637,491	12,096,545	612.94	0.96
Sh-F	1	11,637,491	6,289,822	318.71	1.85
AC	2	52,218,939	28,926,912	162.86	1.80
Huffman	2	52,218,939	27,909,547	157.13	1.87
LZW	2	52,218,939	35,397,098	199.29	1.47
RLE	2	52,218,939	62,153,377	349.93	0.84
Sh-F	2	52,218,939	34,717,441	195.46	1.50

## Data Availability

All used data sources are mentioned in the article.

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
