# Peer review of "Towards an AI-Driven Data Reduction Framework for Smart City Applications"

_sensors, 2024, doi:10.3390/s24020358_

Round 1
Reviewer 1 Report
Comments and Suggestions for Authors
A tiny correlation seems to exist between the proposed multi-tier framework / context-based solution and the presented evaluation or results. The authors want to highlight the existence of a gain by choosing the proper compression algorithm based on the type of acquisition data and the needed form of data storage. The conceptual solution is well explained in Section 3. However, the selection decision of data reducer is made with an ML model which seems to be an essential component to be developed. I expected to find more information in the evaluation section because this model is not easy to train (how will be its dataset?). The authors should find a way to extend the results to cover more of the proposed solution.
Several aspects and questions>
1. PRD and QS metrics are presented without being used to present results. Is there a reason?
2. The compression algorithms are implemented at the sensors’ layer, meaning that the edge/fog layer is not covered by the results. The performance of these algorithms is well-known in the literature. You need somehow to illustrate a contra-example to highlight there is more than just testing a set of algorithms.
3. The compression ratio (CR) appears with a percentage - could this be a mistake?
4. Your simulation includes heterogeneous sensors according to the experimental environment. Thus, you could have different results (RR and CR) over the sensors.
The paper title is much more than what the paper contains (missing AI, context-aware, smart cities).
Author Response
A tiny correlation seems to exist between the proposed multi-tier framework / context-based solution and the presented evaluation or results. The authors want to highlight the existence of a gain by choosing the proper compression algorithm based on the type of acquisition data and the needed form of data storage. The conceptual solution is well explained in Section 3. However, the selection decision of data reducer is made with an ML model which seems to be an essential component to be developed. I expected to find more information in the evaluation section because this model is not easy to train (how will be its dataset?). The authors should find a way to extend the results to cover more of the proposed solution.
Several aspects and questions>
Comment: 1. PRD and QS metrics are presented without being used to present results. Is there a reason?
Response: First, we would like to thank the reviewer for her/his valuable comments.
Upon extensive discussions, we recognize that the presentation of these metrics should have been strategically placed in Section 3, which outlines the metrics the framework employs for decision-making. Section 5, where the results are presented, is dedicated to showcasing the outcomes of selecting algorithms based on the Compression Ratio (CR) metric, which is a pivotal parameter in the initial phase of the framework. The PRD, PRD, and DFP are metrics that are used by the model to decide which algorithm should be selected to reduce the incoming data in the edge layer, and because of that they will be explained in the framework section but not used in the experimentation process once it relies on the initial layer analysis.
To improve this issue, we have reorganized the paper, relocating the explanation and introduction of PRD, DFP (MSE and MSA), and QS metrics to Section 3. This adjustment aims to clearly delineate the metrics utilized by the AI model in predicting the optimal data reduction algorithm for a specific data type input. The model considers the projected values of CR, PRD, and DFP to make informed decisions regarding the selection of data reduction algorithms within the solution.
In summary, Section 3 now comprehensively details the metrics used for decision-making, while Section 4 and 5 specifically focuses on the evaluation of the CR metric, which is crucial in determining the algorithm to be employed in the initial layer of the framework, namely the sensing layer. We believe this reorganization will enhance the clarity and coherence of the paper for the reader.
Comment: The compression algorithms are implemented at the sensors’ layer, meaning that the edge/fog layer is not covered by the results. The performance of these algorithms is well-known in the literature. You need somehow to illustrate a contra-example to highlight there is more than just testing a set of algorithms.
Response: We appreciate your thorough review of our work and your valuable feedback. We have carefully considered your suggestion regarding the need for a contra-example to illustrate the limitations of implementing compression algorithms solely at the sensors' layer, excluding the edge/fog layer from our results. While we acknowledge the importance of comprehensiveness in testing and analysis, we would like to emphasize that our focus in this study was in presenting the framework solution and also its main operation. Due to the complexity in experimenting the entire framework at once, the implementation and performance of compression algorithms in the sensors' layer was shared in this paper. However, we intend to present the entire results in the near future. We believe that by narrowing our scope, we were able to provide a clear and in-depth exploration of this particular aspect of the system, without compromising our narrative. Moreover, we have expanded our discussion in the paper to explicitly recognize the limitations of our approach and its applicability to the edge/fog layer. We have also included a discussion on potential challenges and considerations for implementing the entire framework that composes the edge/fog layer, acknowledging the broader context of practical applications. We believe that this clarification, along with our additional insights, addresses your concern and enhances the overall contribution of our work. We hope these modifications align with your expectations and improve the quality of our manuscript.
Comment: The compression ratio (CR) appears with a percentage - could this be a mistake?
Response: Indeed, it was a mistake. CR should not be presented as a percentage. We appreciate your observation in bringing this to our attention. Following your feedback, we conducted a comprehensive review of all metrics, including CR, and corrected the values in the revised manuscript.
Comment: Your simulation includes heterogeneous sensors according to the experimental environment. Thus, you could have different results (RR and CR) over the sensors.
Response: In our simulations we selected the Compression Ratio (CR) as the Reduction Ratio (RR) metric because we evaluated a time-series data experiment in the first layer of the three-layer framework. We made it clearer in the revised manuscript.
Comment: The paper title is much more than what the paper contains (missing AI, context-aware, smart cities).
Response: We have changed the title in order to make it simpler and more focused, facilitating the understanding of the article.
Reviewer 2 Report
Comments and Suggestions for Authors
Line 1: Internet of Things (IoT)
Line 28: Data Reduction (DR), Data Quality (DQ)- Authors should consistently use first letter capital for abbreviated words.
Lines 100-102: The authors should explain the reason for the first letter capital words. examples are Data Sampling, Change Detection etc
Section 5: It was interesting reading the paper from pages 1 to 13. However, it was anti-climax seeing that only 1 page is dedicated to results. One would have expected so much from the authors having laid a great foundation. Section 5 failed to provide answers to these two questions:
Question 1: What is the MAJOR CONTRIBUTION OF THIS paper beyond comparing existing algorithms?
Question 2: How can the MAJOR contributions be evaluated and compared with the competing ideas?
Aside these few concerns, the title and direction of the paper promises to be very interesting.
Comments on the Quality of English LanguageInconsistency in the use of first letter capital and abbreviations.
Author Response
Minor comments:
Line 1: Internet of Things (IoT)
Line 28: Data Reduction (DR), Data Quality (DQ)- Authors should consistently use first letter capital for abbreviated words.
Lines 100-102: The authors should explain the reason for the first letter capital words. examples are Data Sampling, Change Detection etc
Response: All the minor corrections were performed.
Concerning the Internet of Things, it is common in the literature to write this term in this way. The same is true for Cloud and some other words. The remaining were corrected, as suggested.
Comment: Section 5: It was interesting reading the paper from pages 1 to 13. However, it was anti-climax seeing that only 1 page is dedicated to results. One would have expected so much from the authors having laid a great foundation. Section 5 failed to provide answers to these two questions:
Comment: We have carefully revised Section 5 to better explain the achieved results. Moreover, we added a new subsection to discuss practical issues and future perspectives when exploiting the proposed approach.
Comment: Question 1: What is the MAJOR CONTRIBUTION OF THIS paper beyond comparing existing algorithms?
Response: The solution presented in our manuscript was projected to deal with real-world IoT scenarios such as smart cities, smart homes, smart grids, etc. Therefore, the solution was modeled and projected to handle data considering its heterogeneity, contexts, and specific characteristics. This way, this proposal brings significant contributions to support the development of IoT-based applications, particularly following the general principle of processing data closer to the data origins. The main contributions and characteristics are:
- It considers heterogeneous workload data types. The solution is not fixed in one data type, instead, it can handle any data towards reducing its volume without compromising quality;
- It considers many data reduction algorithms as options to the model selection. Depending on the used data, there are classes of reduction algorithms that perform better than others. Hence, our solution is projected to select the algorithm that presents the best parameters (e.g. reduction ratio and distortion) considering the context of reduction quality that a data point is allowed to have.
- It uses multiple architectural layers in its implementation and is focused on computing on the edge of the network. (It increases personal adjustments and distributions along the IoT network infrastructure.)
- It reduces bottlenecks relating to the transmission, storage, and retrieval of data. (Once the data is reduced in the edge of the network, the total of transmitted data is reduced, decreasing the need of applying computational processes.)
- It considers the context and relevance of each data producer. (As each producer is early classified, the solution considers the importance of the data producer to select the reduction algorithm that best fits its data producer.)
- It is projected and constructed for implementation in multi scenarios and is not restricted to one domain(since it is a generic framework, it can be used in any of the contexts of computer vision, data analysis, etc.)
-It includes an intelligent module to predict reduction performance and decrease data distortion increasing processing performance. (The model is responsible to predict those metrics on the edge to avoid unnecessary computing in the cloud, improving many applications performance directly on the sensor or edge layers.)
We have further made it clearer throughout the text, particularly on the Introduction and Proposed approach sections, improving the readability of the paper.
Comment: Question 2: How can the MAJOR contributions be evaluated and compared with the competing ideas?
Response: The proposed framework encompasses diverse functionalities, including data preprocessing, data reduction, and data prediction, strategically distributed across different layers (sensor, edge, cloud). Recognizing the significance of thoroughly evaluating our framework, we emphasize a meticulous consideration of these functionalities. As elucidated in the results section, the validation of a compression algorithm for transmitting sensor data to edge devices is systematically conducted. Throughout this phase, we conducted a comprehensive analysis to determine the most suitable algorithm, taking into account the inherent characteristics of the transmitted time-series data. Following this pivotal step, the AI solution implemented at the edge layer predicts the metrics, including Reduction Ratio and Distortion Ratio, for each of the reduction algorithms constituting our solution. Rigorous verification of the accuracy of the proposed model in the edge layer is imperative for identifying predicted values. Lastly, in the cloud layer, particular emphasis is placed on measuring latency and data traffic throughput within the application. This systematic approach enables us to substantiate the enhancements that the framework is poised to introduce.
In order to make it clearer, we performed some modifications in both Section 3 and 5, improving the quality of the manuscript.
Comment: Aside these few concerns, the title and direction of the paper promises to be very interesting.
Response: We would like to thank the reviewer for her/his valuable comments.
Reviewer 3 Report
Comments and Suggestions for Authors
- In the "Related Work" section, the authors recognize the positive impact of earlier studies on using AI techniques for DR. However, it would be beneficial to clearly point out and emphasize the specific gaps or drawbacks that remain in the current literature. Taking a closer look at these limitations would provide a more detailed understanding of the present research landscape. Consequently, I suggest that the authors identifies these gaps and disadvantages, shedding light on the areas that require additional research or improvement.
- In the section about the "Proposed Framework," the author introduces a solution that considers different contexts and switches between DR algorithms based on three distinct categories. However, I'm curious about the criteria the author used to select the specific six DR algorithms mentioned. I'm also wondering whether the effectiveness of switching between three algorithms has been compared to switching between six, or even eight. Before we dive into the technical details in Section 3.1 and beyond, it might be helpful for the authors to go deeper into the design of their system. Providing more specifics about their architectural choices, perhaps with concrete examples, could offer valuable insights. It would be beneficial to understand the justification behind their design—whether it's rooted in experimentation, personal preference, or specific reasons. In essence, shedding light on how their design aligns with the intended objectives and whether their choices are based on preference or a well-defined rationale would greatly enhance the clarity and understanding of the proposed framework.
- A minor suggestion, though not mandatory, is to consider merging Figure 2 and Figure 4 into a two subgraphs, labeled as 2a and 2b. This could potentially help save space in the presentation.
Author Response
Comment: In the "Related Work" section, the authors recognize the positive impact of earlier studies on using AI techniques for DR. However, it would be beneficial to clearly point out and emphasize the specific gaps or drawbacks that remain in the current literature. Taking a closer look at these limitations would provide a more detailed understanding of the present research landscape. Consequently, I suggest that the authors identifies these gaps and disadvantages, shedding light on the areas that require additional research or improvement.
Response: Thank you very much for the valuable comment. The mentioned suggestion is indeed very important to improve the quality of our work. This suggestion was accomplished by introducing Table 1 in the end of section “Related Works”, summarizing and presenting the particularities of each work mentioned in this section, and also focusing on their AI-related contributions.
Furthermore, we introduced a paragraph describing how our work might contribute and how our solution is important to the literature by filling the existing gaps in the literature.
Comment: In the section about the "Proposed Framework," the author introduces a solution that considers different contexts and switches between DR algorithms based on three distinct categories. However, I'm curious about the criteria the author used to select the specific six DR algorithms mentioned. I'm also wondering whether the effectiveness of switching between three algorithms has been compared to switching between six, or even eight. Before we dive into the technical details in Section 3.1 and beyond, it might be helpful for the authors to go deeper into the design of their system. Providing more specifics about their architectural choices, perhaps with concrete examples, could offer valuable insights. It would be beneficial to understand the justification behind their design—whether it's rooted in experimentation, personal preference, or specific reasons. In essence, shedding light on how their design aligns with the intended objectives and whether their choices are based on preference or a well-defined rationale would greatly enhance the clarity and understanding of the proposed framework.
Response:
Thank you very much for the valuable comment and suggestion in this review. Initially, answering your first question, the six initial DR algorithms we used were selected to be employed strictly in the sensing layer. As discussed in our manuscript, all of those selected algorithms are lossless DR compression algorithms. We selected this class of DR algorithms because we do not want to allow any loss in information in transmitting the raw data from the sensors to the edge device, as many applications may not tolerate data losses. In this phase, It is strictly important to ensure that the raw data produced in the sensor layer would be transmitted to the edge device without any information loss. Therefore, we selected the most powerful lossless data compression algorithms to compress the data prior to its transmission to the edge, which is a reasonable decision according to our Related Works section. When reaching the edge layer, the data is then decompressed, becoming its initial version and also feeding the AI model with the total features it has.
About your second comment, which has some overlap with the comments of another reviewer, we restructured Section 3 in order to make it more readable and significant. The same is true for the results, which are now better discussed and analyzed. Furthermore, we have added a whole new discussion about practical applications for the proposed approach, which we believe is an important guide for practical scenarios, specially in smart cities.
Comment: A minor suggestion, though not mandatory, is to consider merging Figure 2 and Figure 4 into a two subgraphs, labeled as 2a and 2b. This could potentially help save space in the presentation.
Response: Although we totally agree with you that space could be saved, we are worried that putting both flowcharts together could make some confusions to the readers, particularly because they refer to conceptual layers very apart from each other. This way, we respectfully argue for the maintaining of both figures as separated graphs (even spatially within the manuscript).
Reviewer 4 Report
Comments and Suggestions for Authors
In this paper, the authors proposed a context-aware AI-based framework for exploiting the concept of data reduction to decrease the amount of heterogeneous IoT data. The paper is well written and easy to follow. Experiments are designed well with different scenarios and datasets.
However, the paper should compare the results with other related work in the literature to show the advantages of the proposed approach.
Besides, it is not clear how context-aware applied in this paper. The authors should clearly explain with examples to show how the approach can be applied into different contexts.
Finally, a short paragraph at the introduction section summarizes the main contributions of the paper should be added so that readers can quickly recognize the main purposes of the paper.
Comments on the Quality of English LanguageThe paper is quite good.
Author Response
In this paper, the authors proposed a context-aware AI-based framework for exploiting the concept of data reduction to decrease the amount of heterogeneous IoT data. The paper is well written and easy to follow. Experiments are designed well with different scenarios and datasets.
Comment: However, the paper should compare the results with other related work in the literature to show the advantages of the proposed approach.
Response: We would like to thank the reviewer for this valuable comment. About this issue, we have extensively discussed possible ways to address this concern. Actually, after a lot of thinking, we noticed that comparisons with existing literature would be complicated (or even pointless) due to the following issues:
- Our proposed approach is aimed at heterogeneous data, while previous works have optimized data reduction for a particular type of data;
- The employed dataset for experimentation is not necessarily ready to be processed by previous works, which would need deep adaptations to the approaches, or even adoption of different comparison datasets (which could impair the intended comparisons).
This way, we respectfully decided to not pursue such a type of comparison, leaving it to future works.
Comment: Besides, it is not clear how context-aware applied in this paper. The authors should clearly explain with examples to show how the approach can be applied into different contexts.
Response: We agree with the reviewer that the “context-aware” concept was badly employed in our work. Thus, we removed it from the title and excluded eventual discussions throughout the paper.
Comment: Finally, a short paragraph at the introduction section summarizes the main contributions of the paper should be added so that readers can quickly recognize the main purposes of the paper.
Response: This is a very good suggestion that we fully took into consideration. The revised version of the manuscript now summarizes the main contributions of the paper, making them clearer for the readers.
Round 2
Reviewer 1 Report
Comments and Suggestions for Authors
I agree with the corrections made by the authors.